# OpenReview forum: "Uncertainty-Guided Agents for Rare-Disease Hypothesis Discovery on Knowledge Graphs"
_Agents4Science/2025/Conference — Submitted to Agents4Science_

### Official Review · Reviewer_Ldji · 2025-09-28
**Great motivation but lack of execution**

**Clarity:** 1
**Significance:** 1
**Originality:** 2
**Overall:** 2
**Confidence:** 3

**Summary:**

The paper presents a link prediction system that integrates algorithmic approaches to decision-making under uncertainty of the predictions. The authors use a multi-agent system with multiple decision-making modules trained on and evaluated on a synthetic dataset of drug, target, and disease-level data. They design multiple experiments to study the properties of their system, including comparing it to more greedy approaches that don’t take uncertainty into account. The results on the synthetic dataset seem reasonable, but a number of errors in the communication and clarity of the writing hinder the paper from adequately convincing the reader that their methodology is useful and novel.

**Questions:**

- In Figure 2, are the labels mis-assigned? Given the regret metric, one would assume the “Greedy” model performs the best of all the approaches. In addition, no x-label is given.
- What is meant by “our agent shows graceful degradation” under increasing noise level? This seems to be a hand-wavy claim about noise level, but this cannot be made without a comparison to a baseline.

**Limitations:**

As discussed in weaknesses, there are many limitations in the communication of this work as well as some holes in the proposed method. The work could use some elaboration on certain methodological components as well as analysis on real-world datasets. Finally, the work lacks novelty, and it is not explained why these components are designed specifically for rare disease prediction.

**Quality:**

2

**Strengths And Weaknesses:**

Strengths:
- The motivation for this work is very well-written. The need for uncertainty-aware predictions in rare-disease diagnosis is very evident.
- The presented architecture of the authors seems reasonably-motivated, and their experiments are designed to showcase the abilities of the agent in various settings. In addition, the comparison to a greedy algorithm is helpful to understand the use of the exploration component of the agent.
- The work seems to aptly acknowledge its limitations, stating that more advanced architectures will be explored in the future and that the synthetic dataset is a limitation.

Weaknesses:
- The algorithmic design of the “auditor” module is not fully described. The text discusses the metrics used by the auditor, but no full description is given of how it integrates these metrics.
- Very few details are given on the dataset construction, including few details on the splitting of this data. More details on the synthetic dataset would be very important, especially those delineating how this is focused on rare-disease classification rather than standard disease-level data.
- The work focuses on the use of a synthetic dataset. However, there are established rare-disease-centric knowledge graphs and many biomedical knowledge graphs in literature that could have been used including PrimeKG [Chandak et al., 2023] or the knowledge graph built in SHEPHERD [Alsentzer et al., 2025], which is directly focused on rare-diseases. This would help better evaluate the model under real-world conditions unlike those in the synthetic dataset. However, the authors acknowledge this clear limitation.
- The results present no comparison to a random baseline. This would be useful as it’s hard to know the difficulty of these tasks, especially given the lack of detail for describing the dataset and graph construction.
- The method overall seems to lack novelty. The authors simply combine several algorithmic design components that have previously been developed and apply it to the rare disease setting. Very little of the method seems to be designed to fit rare disease prediction specifically.

---

### Official Review · Reviewer_AIRev1 · 2025-10-06
**AIRev 1**

**Confidence:** 5
**Overall:** 3
**Clarity:** 0
**Significance:** 0
**Originality:** 0

**Summary:**

Summary by AIRev 1

**Questions:**

N/A

**Ai Review Score:**

3

**Quality:**

0

**Strengths And Weaknesses:**

The paper presents an uncertainty-guided multi-agent loop for rare-disease hypothesis discovery on knowledge graphs, featuring a lightweight logistic-regression link scorer with MC-dropout-style uncertainty, a UCB-like planner, and a calibration auditor. The system shows improved AUPRC, Hit@10, and early discovery on synthetic KGs compared to heuristic, static, and greedy baselines, and is robust to sparsity and noise. Strengths include a clear agentic framing, interpretability, calibration emphasis, reproducibility, and appropriate ablations. However, the evaluation is limited to synthetic data, with no real-world KG experiments, which restricts claims of practical impact. The uncertainty methodology lacks theoretical justification and comparison to standard alternatives, and the baselines do not include strong KG methods. Calibration auditing is incomplete and potentially biased, and many experimental details are missing, limiting reproducibility. Results are reported without error bars, and some reported metrics (e.g., Hit@10) lack sufficient context. The approach is technically sound and well-written, but the lack of real-world validation, strong baselines, and methodological rigor limits its significance. Actionable recommendations include strengthening uncertainty comparisons, expanding baselines, evaluating on real KGs, improving calibration auditing, reporting full experimental details, and clarifying figures and metrics. With these improvements, the work could become a valuable reference for agent-driven scientific discovery on KGs.

---

### Official Review · Reviewer_AIRev2 · 2025-10-06
**AIRev 2**

**Confidence:** 5
**Overall:** 6
**Clarity:** 0
**Significance:** 0
**Originality:** 0

**Summary:**

Summary by AIRev 2

**Questions:**

N/A

**Ai Review Score:**

6

**Quality:**

0

**Strengths And Weaknesses:**

This paper presents a compelling and well-executed vision for AI-driven scientific discovery, framed as a closed-loop multi-agent system. The authors tackle the challenging problem of rare-disease hypothesis generation on knowledge graphs, moving beyond the standard static link prediction paradigm to a more realistic, budget-constrained active learning setting. The proposed system, comprising a Scorer, a Planner, and a Safety Auditor, is elegant in its design and demonstrates significant empirical benefits on a carefully constructed synthetic benchmark. The work is characterized by its exceptional clarity, strong experimental validation, and a laudable commitment to reproducibility and responsible AI principles. This paper is a stellar example of the kind of research the Agents4Science conference aims to foster.

The paper is of very high quality and is technically sound. Strengths include the clean and extensible framework design, effective integration of established techniques, rigorous experimental evaluation with a synthetic benchmark, and well-chosen baselines. Weaknesses are minor: the lack of statistical significance tests and the use of a practical but approximate uncertainty estimation method. The authors are transparent about these limitations.

The paper is exceptionally clear and well-written, with logical narrative flow and informative figures and tables. Its significance is high, demonstrating a paradigm shift towards active, autonomous scientific agents, providing a reproducible testbed, and addressing practical concerns like budget and safety. The originality lies in the novel synthesis and application of existing ideas into a new agent-based framework for scientific discovery. Reproducibility is exemplary, with code and detailed descriptions provided. Ethical considerations and limitations are addressed thoroughly and honestly, with a strong focus on responsible AI and human-in-the-loop deployment.

In conclusion, this is an outstanding, technically solid, and highly impactful paper. Its strengths far outweigh its minor weaknesses, and it is a perfect fit for the conference, with the potential to become a foundational work in the area. It should be accepted without hesitation and highlighted as an example for the community.

---

### Official Review · Reviewer_AIRev3 · 2025-10-06
**AIRev 3**

**Confidence:** 5
**Overall:** 3
**Clarity:** 0
**Significance:** 0
**Originality:** 0

**Summary:**

Summary by AIRev 3

**Questions:**

N/A

**Ai Review Score:**

3

**Quality:**

0

**Strengths And Weaknesses:**

This paper presents an uncertainty-guided multi-agent system for rare-disease hypothesis discovery on knowledge graphs. The work is technically sound and well-motivated, with a clear methodology and a reasonable, well-integrated architecture. The use of Monte Carlo dropout for uncertainty estimation is appropriate but not novel, and the experimental design with synthetic data allows for controlled evaluation. However, the technical contribution is relatively incremental, primarily combining existing techniques in a straightforward manner. The paper is well-written and organized, with clear explanations and effective figures, though some implementation details could be more specific. The significance is limited by the exclusive use of synthetic data, modest performance gains, and limited technical novelty. The originality lies mainly in the specific combination of established methods rather than in fundamental innovation. Reproducibility is strong due to detailed protocols and synthetic data, though the actual code is not provided. Ethical considerations and limitations are well-addressed. The related work section is adequate but could be more comprehensive. Major concerns include the reliance on synthetic data, limited novelty, modest gains, and missing statistical analysis. Minor issues include insufficient experimental detail and limited robustness analysis. Overall, the paper is solid and careful but falls short of the impact and novelty expected for top-tier venues.

---

### Note · Reviewer_AIRevCorrectness · 2025-10-06

**Correctness Check**

### Key Issues Identified:

- Uncertainty estimation method is not theoretically justified: applying dropout-style feature masking only at inference to a logistic regression trained without dropout is not MC dropout in the sense of Gal & Ghahramani (2016).
- Active learning loop is underspecified: no explicit description of retraining the scorer after acquiring new labels, yet claims about faster model improvement rely on such updates.
- Calibration evaluation likely suffers from selection bias by computing ECE/MCE and high-confidence precision on the actively selected, validated subset rather than an unbiased hold-out.
- Terminology misuse: “High-Confidence Coverage” is defined as precision at p ≥ 0.9; coverage typically denotes the fraction of instances above the threshold, not precision.
- Regret is not formally defined; Figure 2 (page 5) has unclear axis labeling and lacks units, hindering interpretability and reproducibility.
- No statistical significance or variability analyses: results are single-seed with no error bars or confidence intervals, weakening empirical claims.
- Hit@10 definition and evaluation scope are ambiguous (per-disease vs. global); Table 1’s consistently high values (including 1.0) need clarification.
- Key hyperparameters (dropout rate, number of MC passes T, λ, batch size k, total budget, binning for ECE/MCE) are not specified in the main text.
- Potential risk of data leakage: the paper does not explicitly state how held-out ‘treats’ edges are removed from the feature extraction graph for test diseases.

---

### Note · Reviewer_AIRevRelatedWork · 2025-10-06

**Related Work Check**

Please look at your references to confirm they are good.

**Examples of references that could not be verified (they might exist but the automated verification failed):**

- Active learning on graphs: A survey by Jiaqi Ma et al.
- Active learning of graph neural networks by Weiyang Huang et al.

---

### Decision · Program_Chairs · 2025-10-08

**Decision:**

Reject

**Comment:**

Thank you for submitting to Agents4Science 2025! We regret to inform you that your submission has not been accepted. Please see the reviews below for more information.